# Obatoclax, a Pan-BCL-2 Inhibitor, Downregulates Survivin to Induce Apoptosis in Human Colorectal Carcinoma Cells Via Suppressing WNT/β-catenin Signaling

**DOI:** 10.3390/ijms21051773

**Published:** 2020-03-05

**Authors:** Chi-Hung R. Or, Chiao-Wen Huang, Ching-Chin Chang, You-Chen Lai, Yi-Ju Chen, Chia-Che Chang

**Affiliations:** 1Department of Anesthesiology, Tungs’ Taichung MetroHarbor Hospital, Taichung 43503, Taiwan; richardor92@yahoo.com.tw; 2Institute of Biomedical Sciences, National Chung Hsing University, Taichung 40227, Taiwan; minisky1214@hotmail.com (C.-W.H.); za12323123@gmail.com (C.-C.C.); z091938882747@gmail.com (Y.-C.L.); 3Department of Dermatology, Taichung Veterans General Hospital, Taichung 40705, Taiwan; 4Department of Medicine, National Yang Ming University, Taipei 11221, Taiwan; 5Department of Life Sciences, The iEGG and Animal Biotechnology Research Center; Ph.D. Program in Translational Medicine, Rong Hsing Research Center for Translational Medicine, National Chung Hsing University, Taichung 40227, Taiwan; 6Traditional Herbal Medicine Research Center, Taipei Medical University Hospital, Taipei 11031, Taiwan; 7Department of Medical Research, China Medical University Hospital, Taichung 40447, Taiwan; 8Department of Biotechnology, Asia University, Taichung 41354, Taiwan

**Keywords:** obatoclax, survivin, WNT/β-catenin, apoptosis, colorectal cancer

## Abstract

Colorectal cancer (CRC) is a highly prevailing cancer and the fourth leading cause of cancer mortality worldwide. Aberrant expression of antiapoptotic BCL-2 family proteins is closely linked to neoplastic progression and chemoresistance. Obatoclax is a clinically developed drug, which binds antiapoptotic BCL-2, BCL-xL, and MCL-1 for inhibition to elicit apoptosis. Survivin is an antiapoptotic protein, whose upregulation correlates with pathogenesis, therapeutic resistance, and poor prognosis in CRC. Herein, we provide the first evidence delineating the functional linkage between Obatoclax and survivin in the context of human CRC cells. In detail, Obatoclax was found to markedly downregulate survivin. This downregulation was mainly achieved via transcriptional repression, as Obatoclax lowered the levels of both *survivin* mRNA and promoter activity, while blocking proteasomal degradation failed to prevent survivin from downregulation by Obatoclax. Notably, ectopic survivin expression curtailed Obatoclax-induced apoptosis and cytotoxicity, confirming an essential role of survivin downregulation in Obatoclax-elicited anti-CRC effect. Moreover, Obatoclax was found to repress hyperactive WNT/β-catenin signaling activity commonly present in human CRC cells, and, markedly, ectopic expression of dominant-active β-catenin mutant rescued the levels of survivin along with elevated cell viability. We further revealed that, depending on the cell context, Obatoclax suppresses WNT/β-catenin signaling in HCT 116 cells likely via inducing β-catenin destabilization, or by downregulating LEF1 in DLD-1 cells. Collectively, we for the first time define survivin downregulation as a novel, pro-apoptotic mechanism of Obatoclax as a consequence of Obatocalx acting as an antagonist to WNT/β-catenin signaling.

## 1. Introduction

Colorectal cancer (CRC) is the second and third most common cancer diagnosed in women and men, respectively, and is the fourth leading cause of cancer death worldwide [1]. Approximately 30% to 50% of CRC patients will develop liver metastases [2]. Surgery, chemotherapy, targeted therapy, radiation therapy, and immunotherapy are the current options for CRC treatment. Surgery alone can be enough to treat stage I and stage II CRC, whereas systemic chemotherapy remains as the primary treatment in stage III and IV. The five-year relative survival rate for CRC patients is 65%, ranging from 91% and 82% for patients in stage I and II, respectively, to 12% for stage IV disease [3].

Dysregulation of signaling pathways including EGFR/MAPK, Notch, PI3K/AKT, TGF-β, or WNT/β-catenin has been linked to CRC genesis and progression [4]. In particular, WNT/β-catenin signaling is prominent among these pathways, given its aberrant activation is present in almost all CRC cases [5]. Inactivated mutations in the *APC* gene or activating mutations in the β-catenin-encoding gene *CTNNB1* together account for the majority of genetic lesions in CRC cells, which lead to stabilization and ensuing nuclear translocation of β-catenin to facilitate TCF/LEF-dependent transcription of WNT/β-catenin signaling target genes to drive cell proliferation, metastasis, and cancer stemness [6,7,8]. It is generally believed that hyperactive β-catenin-mediated transcriptional activation underlies the initiation and malignant progression of CRC; accordingly, components in the WNT/β-catenin signaling pathway represent promising molecular targets for CRC therapeutics [6,9].

Survivin, a well-defined WNT/β-catenin target gene [10], is the smallest member of the inhibitor of apoptosis (IAP) protein family and, of note, is the fourth most elevated mRNA in the human cancer transcriptome while it is barely detected in normal adult cells [11,12]. Functionally, survivin is essential for mitosis, particularly during the metaphase–anaphase transition, acts as an apoptosis inhibitor, and promotes cell migration, angiogenesis, and cancer stemness maintenance. As expected, survivin upregulation is highly associated with pathogenesis, resistance to chemo- and radiotherapies, and poor prognosis for a variety of human malignancies, including CRC [12,13,14,15,16,17,18]. Hence, considering the cancer-selective expression pattern and pivotal role of survivin in cancer pathogenesis, targeting survivin represents a promising approach for developing novel cancer therapeutics [19,20].

Obatoclax, a synthetic derivative of bacterial prodiginines [21], is a clinically developed small-molecule pan-BCL-2 inhibitor that functions by blocking BH3-mediated binding of BH3-only proteins or BAX/BAK to antiapoptotic BCL-2, BCL-xL, and MCL-1, causing BAX/BAK activation to trigger apoptosis [22]. Phase I/II clinical trials have revealed the anticancer potential of Obatoclax as a single agent or in combination with other chemo- and radiation therapies [23,24,25]. Notably, given overexpression of antiapoptotic BCL-2 family members is closely linked to therapeutic resistance in cancer cells, Obatoclax has been demonstrated to facilitate drug sensitization of chemoresistant cells in both hematological [23] and solid tumors [26]. In addition to eliciting BAX/BAK-dependent apoptosis, Obatoclax is able to provoke cell death in BAX/BAK-deficient cancer cells [27]. To this end, Obatoclax is known to provoke autophagic cell death [28,29,30] or necroptosis [31] in different cell systems. In view of that, a better molecular understanding about Obatoclax-induced cytotoxicity is fundamental to apply Obatoclax to cancer treatment, either as a single agent or in combination with other cancer therapeutics.

Herein, we reported the first evidence establishing the molecular connection among Obatoclax, survivin, and WNT/β-catenin signaling in the context of CRC cell lines. We proved that, aside from acting as a pan-BCL-2 inhibitor, Obatoclax’s proapoptotic action involves survivin downregulation via suppressing WNT/β-catenin signaling. Our novel discovery thus reveals the multiple modes of Obatoclax’s pharmacological action but also the potential application of Obatoclax to CRC therapy.

## 2. Results

### 2.1. Obatoclax Is Cytotoxic and Proapoptotic against Multiple Human Colorectal Carcinoma Cell Lines

To examine the possible anti-CRC effect of Obatoclax, a panel of human colorectal carcinoma cell lines including DLD-1, HCT 116, LoVo, and WiDr were examined for cell viability after 48 h treatment with Obatoclax. A dose-dependent reduction in cell viability of all Obatoclax-treated human CRC cells was observed, with IC_50_ values of 257.19 ± 1.46, 89.96 ± 1.68, 283.82 ± 3.46, and 231.04 ± 2.01 nM for DLD-1, HCT 116, LoVo, and WiDr cells, respectively (Figure 1A). In addition, the clonogenicty of all cell lines was dropped to about 40% of drug-free control when treated with 200 nM of Obatoclax (*p* < 0.001) (Figure 1B). Thus, these results validated the in vitro cytotoxicity of Obatoclax against all human CRC cell lines examined. Next, we addressed whether apoptotic death underlies the nature of Obatoclax-induced cytotoxicity. As shown in Figure 1C, Obatoclax treatment led to a dose-dependent increase in the levels of cleaved poly(ADP-ribose) polymerase (PARP), confirming caspase activation and hence apoptosis induction. Moreover, we noticed that in all Obatoclax-treated cancer cells, the levels of cleaved caspases 8, 9, and 3 were elevated in a dose-dependent manner, suggesting that Obatoclax induced apoptosis through engaging both extrinsic and intrinsic apoptotic signaling pathways (Figure 1C). Taken together, these lines of evidence confirmed the cytotoxic and proapoptotic action of Obatoclax on various human colorectal carcinoma cell lines, illustrating the potential anti-CRC effect of Obatoclax.

### 2.2. Obatoclax Downregulates Survivin Primarily at the Level of Transcription

Given survivin’s antiapoptotic and oncogenic roles in CRC, we went on to explore the effect of Obatoclax on survivin in the context of CRC cells. It is noteworthy that survivin protein levels in all tested human CRC cell lines were markedly diminished by Obatoclax in a dose-dependent manner (Figure 2A). To inquire whether Obatoclax downregulates survivin at the level of transcription, we determined the levels of *survivin* mRNA in Obatoclax-treated cells. It is clearly noted that Obatoclax treatment caused a dose-dependent reduction in *survivin* mRNA expression (Figure 2B). Likewise, the human *survivin* promoter activity was dose-dependently lowered upon Obatoclax stimulation (Figure 2C), confirming that Obatoclax repressed the transcription of the *survivin* gene to down-regulate survivin. To further address whether Obatoclax downregulates survivin through promoting survivin protein destabilization, cells were treated with Obatoclax without or with MG132 to block proteasome-mediated degradation of survivin. It is noteworthy that in all human CRC cell lines examined, a blockade of proteasomal degradation failed to nullify Obatoclax-induced survivin downregulation, thus excluding the involvement of survivin protein destabilization in the inhibitory effect of Obatoclax on survivin expression (Figure 2D). Collectively, these data revealed that Obatoclax downregulates survivin in CRC cell lines mainly through inhibiting *survivin* transcription.

### 2.3. Survivin Downregulation is Fundamental to Obatoclax to Elicit Colorectal Cancer Cell Death

The functional significance of survivin downregulation in Obatoclax-mediated anti-CRC action was next addressed. To this end, we established stable clones of DLD-1 and HCT 116 cells with ectopic expression of HA-tagged survivin (HA-survivin) to withstand Obatoclax-induced survivin downregulation. It is noticeable that in both DLD-1 and HCT 116 stable clones, HA-survivin ectopic expression allayed Obatoclax-induced PARP cleavage (Figure 3A). Likewise, compared to their vector controls, the viability of both DLD-1 and HCT 116 HA-survivin stable clones was enhanced (Figure 3B) along with increased levels of clonogenicity (Figure 3C) after Obatoclax treatment. Altogether, these lines of evidence support the notion that survivin downregulation is one of the integral mechanisms of action whereby Obatoclax induces cytotoxicity against human CRC cells.

### 2.4. Obatoclax Inhibits WNT/β-Catenin Signaling in Human Colorectal Carcinoma Cell Lines

With the essential role of survivin downregulation in Obatoclax-mediated anti-CRC action clearly established, we then aimed to elucidate the upstream signaling pathways responsible for Obatoclax-elicited survivin downregulation. In particular, WNT/β-catenin signaling was our main target because of its involvement in promoting *survivin* transcription but also the prerequisite role of its aberrant activation in CRC tumorigenesis. To this end, we evaluated the effect of Obatoclax on the activity of WNT/β-catenin signaling revealed by the cell-based TOPFlash reporter system [32]. It is noteworthy that WNT/β-catenin signaling activity was markedly dropped in DLD-1, HCT 116, LoVo, and WiDr cells upon treatment with 200 nM of Obatoclax (*p* < 0.001) (Figure 4A). To further substantiate the inhibitory effect of Obatoclax on WNT/β-catenin signaling, we performed immunoblotting to assess the expression levels of c-MYC and cyclin D1, both of which are canonical downstream target genes of the WNT/β-catenin signaling pathway [33,34]. As clearly shown in Figure 4B, Obatoclax induced a dose-dependent downregulation of both c-MYC and cyclin D1. Accordingly, these results identified Obatoclax as a potent inhibitor of WNT/β-catenin signaling.

### 2.5. Suppression of WNT/β-Catenin Signaling is Required for Obatoclax to Downregulate Survivin

We next delineated the functional significance of WNT/β-catenin signaling in Obatoclax-induced survivin downregulation and consequent CRC cell death. To this end, we generated DLD-1 and HCT 116 clones with stable expression of a dominant-active β-catenin mutant (∆N90-β-catenin, encoding β-catenin with a deletion of its N-terminal 90 amino acids) to cope with Obatoclax-evoked inhibition of β-catenin-mediated transcriptional activity. It is noteworthy that, in contrast to the vector controls where survivin was markedly downregulated by Obatoclax, ectopic ∆N90-β-catenin expression sustained the levels of survivin, along with attenuation in PARP cleavage (Figure 5A). Moreover, Obatoclax-induced reduction in cell viability and clonogenicity were both markedly rescued in ∆N90-β-catenin stable clones of DLD-1 and HCT 116 cells (Figure 5B,C). Overall, these results underpinned an essential role of WNT/β-catenin signaling blockage in Obatoclax-elicited survivin downregulation, leading to the induction of CRC cell apoptosis.

### 2.6. Obatoclax Thwarts WNT/β-Catenin Signaling Via Cell Context-Dependent Mechanisms

Lastly, the mechanisms by which Obatoclax inhibits WNT/β-catenin signaling were investigated. With respect to β-catenin itself, earlier immunoblotting revealed that β-catenin levels were dose-dependently lessened in Obatoclax-treated HCT 116 cells (Figure 4B); notably, this β-catenin downregulation was likely due to Obatoclax-promoted proteasomal degradation of β-catenin protein, as evidenced by the marked rescue of β-catenin expression when co-treated with MG132 (Figure 6A). In contrast, β-catenin levels remained constant in Obatoclax-treated DLD-1 cells (Figure 4B).

Given activation of WNT/β-catenin signaling culminated at β-catenin nuclear translocation to facilitate transcription factors of the TCF/LEF family, such as TCF4 and LEF1, to initiate transcription of downstream target genes, we probed the effect of Obatoclax on TCF4 and LEF-1 in DLD-1 and HCT 116 cells. It was noticed that, in DLD-1 cells, Obatoclax barely altered TCF4 expression while evidently inducing a dose-dependent downregulation of LEF-1 (Figure 6B, right panel). For HCT 116 cells, a slight decrease in TCF4 levels was induced by 200 nM of Obatoclax, and no LEF-1 expression was observed (Figure 6B, left panel). Taken together, it seems that Obatoclax impedes WNT/β-catenin signaling using distinct modes of action dependent on the context of cell lines. In detail, Obatoclax inhibits WNT/β-catenin signaling in DLD-1 cells likely through downregulating LEF1. For HCT 116 cells, it is reasonable to speculate that Obatoclax downregulates β-catenin to dampen WNT/β-catenin signaling.

## 3. Discussion

In this study, we provide evidence supporting the notion that suppression of WNT/β-catenin signaling-dependent transcriptional activation of survivin is a vital mechanism underlying the proapoptotic effect of Obatoclax on human CRC cells. Specifically, we began to demonstrate that Obatoclax transcriptionally downregulated survivin in multiple human CRC cell lines, illustrating survivin downregulation is a general mode of action of Obatoclax (Figure 1 and Figure 2). Notably, ectopic survivin expression markedly antagonized Obatoclax-induced apoptosis in human CRC cells, confirming the pivotal role of survivin downregulation in Obatoclax-elicited apoptosis (Figure 3). Furthermore, we validated that Obatoclax-induced transcriptional suppression of survivin and resultant apoptosis is likely due to the inhibitory effect of Obatoclax on the hyperactive β-catenin-mediated signaling commonly present in human CRC cells, as evidenced by the abrogation of survivin downregulation and apoptosis induction in Obatoclax-treated CRC cells stably expressing dominant-active β-catenin mutant (Figure 4 and Figure 5). Finally, we revealed a cell context-dependent mode of action of Obatoclax to suppress WNT/β-catenin signaling, either by promoting β-catenin proteasomal degradation operated in HCT 116 cells or by inducing downregulation of LEF-1 as observed in DLD-1 cells (Figure 6). To our best knowledge, the finding about Obatoclax’s inhibitory effect on the aberrantly active β-catenin signaling to provoke survivin downregulation and consequent induction of apoptosis has never been previously documented.

Obatoclax was originally described as a BH3 mimetic, which functions by binding antiapoptotic BCL-2 family proteins for displacing bound BH3-only proteins to initiate BAX/BAK-dependent apoptosis in cancer cells [35]. Intriguingly, growing evidence has unveiled additional ‘off-target’ mechanisms of action of Obatoclax to exert its anticancer effect. For example, Obatoclax downregulated transcriptional repressor YY1 to upregulate death receptor 5 (DR5), leading to TRAIL sensitization [36]. Furthermore, we and others have identified cell-cycle regulator cyclin D1 as a key target lowered by Obatoclax to restrict cell proliferation [37,38]. Along this line, in this report we, for the first time, identified survivin as a novel molecule targeted by Obatoclax to induce apoptosis. Notably, given both survivin and antiapoptotic BCL-2 proteins represent two integral branches of pro-survival mechanisms that promote therapeutic resistance of cancer cells, our discovery that Obatoclax downregulates survivin aside from inhibiting anti-apoptotic BCL-2 proteins is clinically attractive regarding the use of Obatoclax to facilitate drug sensitization in cancer cells with inherent or acquired resistance to cancer therapeutics [16,26,39,40].

Data presented here indicated that Obatoclax downregulates survivin mainly at the level of transcription (Figure 4), which led us to reveal for the first time that Obatoclax suppresses the WNT/β-catenin signaling pathway and consequently induces transcriptional repression of survivin (Figure 5). It is remarkable that mounting preclinical and clinical evidence has linked aberrant activation of WNT/β-catenin signaling to the genesis, progression, stemness maintenance, and drug resistance of a broad spectrum of human cancers, particularly CRC, where over 90% of CRC patients carry activating or loss-of-function mutations in genes of at least one component of the WNT/β-catenin signaling pathway, leading to induced transcription of WNT target genes such as survivin [10,41]. Accordingly, our finding that Obatoclax suppresses WNT/β-catenin signaling activity raises the potential of including Obatoclax in the therapy regimens of CRC. Moreover, it is noteworthy that prodigiosin, a bacterial red pigment with similar chemical structure to Obatoclax, was reported to exert anti-breast cancer effects through inhibition of WNT/β-catenin signaling [42]. The notion that both prodigiosin and its analog Obatoclax are WNT/β-catenin signaling inhibitors suggests that the two pyrrole moieties shared by these two molecules could be the structural basis underlying the inhibitory effect on WNT/β-catenin signaling, but also implicates the possibility that Obatoclax would impair the WNT/β-catenin signaling pathway in cancer cells other than CRC, an issue worthy to be explored in the near future.

Contrary to other human cancers, in CRC almost all mutations that activate WNT/β-catenin signaling are present in genes such as *APC*, *AXIN* or *CTNB1* (encoding β-catenin), resulting in constitutively active β-catenin to promote TCF/LEF-dependent transcription of WNT target genes irrespective of upstream WNT signals [6]. To this end, agents targeting aberrantly active WNT/β-catenin signaling at the level of β-catenin-TCF/LEF-dependent transcription is particularly important for acting as CRC therapeutics. In line with this, our data revealed that Obatoclax thwarts β-catenin-dependent transcription likely through inducing β-catenin destabilization in HCT 116 cells, which carry activating mutation in *CTNB1* but wild-type *APC* [43], or by provoking LEF1 downregulation in DLD-1 cells, which express normal β-catenin but truncated APC [44] (Figure 6). Currently, how Obatoclax disrupts β-catenin-dependent transcription in a cell context-dependent manner remains elusive and warrants further investigation.

In conclusion, we herein prove that, in addition to acting as a pan-BCL-2 inhibitor, Obatoclax triggers cancer cell apoptosis through an additional mechanism of action that involves suppression of WNT/β-catenin signaling to downregulate survivin (Figure 7). Moreover, the identification of Obatoclax as a novel inhibitor of WNT/β-catenin signaling not only underscores the multitarget pharmacological nature of Obatoclax, but also implicates the translation of using Obatoclax to counteract WNT/β-catenin signaling-mediated pathological effects with respect to promoting malignant progression, therapeutic resistance, and cancer stemness of cancer cells.

## 4. Materials and Methods

### 4.1. Chemicals

Obatoclax was purchased from Selleck Chemicals (Houston, TX, USA), prepared as a 20 mM stock solution in dimethyl sulfoxide (DMSO) (Sigma-Aldrich; St. Louis, MO, USA), and stored at 4°C until use. MG132 was obtained from AdooQ BioScience (Irvine, CA, USA). Polybrene was obtained from Sigma-Aldrich (St. Louis, MO, USA).

### 4.2. Cell Culture

Human colorectal carcinoma (CRC) cell lines DLD-1 (ATCC^®^ CCL-221™), HCT 116 (ATCC^®^ CCL-247™), LoVo (ATCC^®^ CCL-229™), and WiDr (ATCC^®^ CCL-218™) were purchased form Bioresource Collection and Research Center (BCRC) (Hsinchu, TWN) and were cultured in RPMI-1640, McCoy’s 5a, F-12K, and MEM medium (Gibco Life Technologies; Carlsbad, CA, USA), respectively. All culture media were supplemented with 10% fetal bovine serum, 1% penicillin–streptomycin, and 1% sodium pyruvate (Gibco Life Technologies). Cells were grown at 37 °C in a humidified environment with 5% CO_2_.

### 4.3. Cytotoxicity Assay

The cytotoxic effect of Obatoclax on CRC cell lines was evaluated by measuring the viability of cells after 24 and 48 h treatment with Obatoclax using CellTiter 96^®^ AQueous One Solution Cell Proliferation Assay (MTS) assay (Promega; Madison, WI, USA), as well as evaluating colony-formation capacity (clonogenicity) of Obatoclax-treated cells. Both cell viability and clonogenicity assays were performed in accordance to our established protocol [45,46].

### 4.4. Quantitative Real-Time Reverse Transcription Polymerase Chain Reaction

The levels of *survivin* mRNA in Obatoclax-treated CRC cells were determined using quantitative real-time reverse transcription polymerase chain reaction (PCR). Total RNA extraction, reverse transcription, and real-time PCR analysis for *survivin* mRNA expression levels were conducted according to our laboratory protocol as previously described [45].

### 4.5. Luciferase Reporter Plasmids

pSRVN-Luc, the luciferase reporter vector for the activity of the human *survivin* promoter covering the region between nucleotides 1824 and 2800 of the human *survivin* gene (GenBank accession number U75285), has been described previously [45]. M50 Super 8x TOPFlash (Addgene plasmid # 12456; http://n2t.net/addgene:12456; RRID:Addgene_12456), the luciferase reporter vector for TCF/LEF-mediated transcriptional activity, and its negative control M51 Super 8x FOPFlash (TOPFlash mutant) (Addgene plasmid # 12457; http://n2t.net/addgene:12457; RRID:Addgene_12457) were gifts from Professor Randall Moon.

### 4.6. Dual-Luciferase Reporter Assay

The levels of the human *survivin* promoter activity as well as the WNT/β-catenin signaling activity revealed by the TOPFlash reporter system were determined by the activity of the respective luciferase reporter. CRC cells (2 × 10^5^) seeded onto 6-well plates were transiently transfected by jetPEI™ (Polyplus; Illkirch, FRA) for 24 h with pSRVN-Luc, M50 Super 8x TOPFlash, or M51 Super 8x FOPFlash, together with a plasmid expressing *Renilla* luciferase (Promega; Madison, WI, USA) for normalization of transfection efficiency. CRC cells transfected with luciferase reporter vectors were then treated with Obatoclax for 24 h, followed by dual luciferase activity assay in accordance to our experimental protocol reported previously [45].

### 4.7. Construction of pBabe-Based Expressing Plasmids for Generating Stable Clones of HA-Survivin or HA-∆N90-β-Catenin

The open reading frames (ORFs) of survivin and the dominant-active β-catenin mutant (deletion of 90 amino acids from the N-terminus of β-catenin; ∆N90-β-catenin) were PCR-amplified from the cDNA pools of the DLD-1 cells as the template using the following primers: survivin—forward: 5′-GGTGC-CCCGACGTTGCCCCCTGCCTG-3′, survivin—reverse: 5′-TCAATCCATGGCAGCCAGCT-GCTCG-3′, ∆N90-β-catenin—forward: 5′-GCTCAGAGGGTACGAGCTGCTATG-3′, and ∆N90-β-catenin—reverse: 5′-CAGCTGCACAGGTGACCAC-3′. All PCR products were first TA-cloned into the pGEM-Teasy vector (Promega; Madison, WI, USA) to verify sequence, followed by directional subcloning to the pBabe-HA vector, a pBabe.puro vector engineered to encode an in-frame N-terminal hemagglutinin (HA) epitope. These pBabe-based expression plasmids were then subjected to production of retroviral particles and subsequent infection to target cells according the established protocols in our laboratory [46].

### 4.8. Immunoblotting

CRC cells (4 × 10^5^) were treated with Obatoclax (0, 100, 200 μM) for 24 h, or treated with Obatoclax for 22 h and then combined with MG132 (20 μM) for an additional 2 h. After drug treatment, cells were lysed, subjected to SDS-PAGE, blot transfer, incubation with primary and secondary antibodies, exposure to enhanced chemiluminescence (ECL) reagent, and final visualization of the antigen-of-interest using the LAS3000 system (Fujifilm; Tokyo, JPN) as stated previously [45,46]. Densitometry analysis of immunoblot signals was accomplished by ImageJ software (National Institute of Health, Bethesda, MD, USA). Anti-survivin (#ab469) and anti-TCF4 (#ab76151) antibodies were bought from Abcam (Cambridge, GBR). Anti-β-catenin (sc-7963) and anti-c-MYC (sc-40) were purchased from Santa Cruz Biotechnology (Dallas, TX, USA). Anti-LEF1 (GTX129186) and anti-GAPDH (GTX627408) were obtained from GeneTex (Hsinchu, TWN), and anti-β-tubulin (#T2200) was bought from Sigma-Aldrich (St. Louis, MO, USA). Primary antibodies against cyclin D1 (#2926), HA-tag (#3724), and cleaved forms of caspases-3 (#9664), -8 (#9496), -9 (#9501), and PARP (#9541) were all purchased from Cell signaling Technology (Boston, MA, USA).

### 4.9. Statistical Analysis

All data were derived from three individual experiments and were expressed as means ± SD. Statistical differences between two independent experimental groups were analyzed using unpaired two-tailed *t* tests. *p* < 0.05 was recognized as statistically significant.

## Figures and Tables

**Figure 1 ijms-21-01773-f001:**
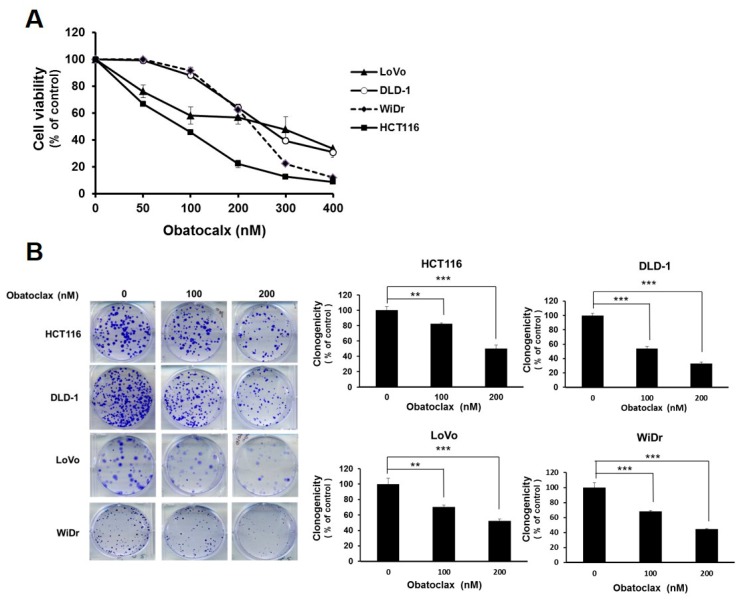
Obatoclax is cytotoxic to human colorectal carcinoma (CRC) cell lines. (**A**) Obatoclax suppresses CRC cell viability. CRC cell lines DLD-1, HCT 116, LoVo, and WiDr were treated with 0~400 nM of Obatoclax for 48 h, followed by cell viability analysis using MTS assay. (**B**) Obatoclax blocks the clonogenicity of CRC cells. CRC cells (2 × 10^2^) after Obatoclax treatment for 24 h were seeded onto 6-well plates and grown in drug-free culture media for 10 days to form colonies. **: *p* < 0.01; ***: *p* < 0.001. (**C**) Obatoclax triggers CRC cell apoptosis. CRC cells were treated with 0~200 nM of Obatoclax for 24 h and then subjected to immunoblot analysis for the status of caspase activation revealed by the levels of cleaved poly(ADP-ribose) polymerase (PARP) (c-PARP), caspase 8 (c-casp 8), caspase 9 (c-casp 9), and caspase 3 (c-casp 3). β-tubulin was used as the equal loading control.

**Figure 2 ijms-21-01773-f002:**
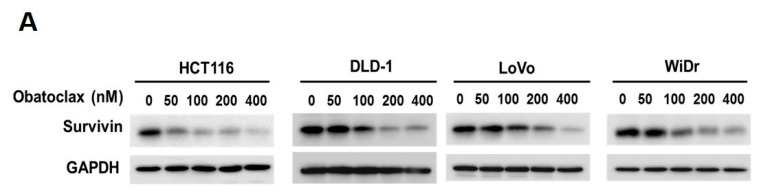
Obatoclax downregulates survivin mainly through transcriptional repression of the *survivin* gene. (**A**) Obatoclax downregulates survivin in CRC cells. DLD-1, HCT 116, LoVo, and WiDr cells were treated with Obatoclax (0~400 nM) for 24 h and then subjected to survivin immunoblotting. Glyceraldehyde 3-phosphate dehydrogenase (GAPDH) levels were used as the equal loading control. (**B**) Obatoclax lowers *survivin* mRNA levels. CRC cells were treated for 24 h with Obatoclax (0, 100, 200 nM), followed by total RNA extraction, reverse transcription, and real-time PCR for the levels of *survivin* mRNA expression. The mRNA levels of TATA box-binding protein (TBP) were used to normalize *survivin* mRNA expression. (**C**) Obatoclax represses the *survivin* promoter activity. CRC cells transiently transfected with pSRVN-Luc (the luciferase reporter construct for the human *survivin* promoter activity) were treated with Obatoclax (0, 100, 200 nM) for 24 h, and then the activity of firefly luciferase was determined thereafter. (**D**) Obatoclax downregulates survivin irrespective to blockage of proteasomal degradation. CRC cells were treated with Obatoclax (200 nM) for 24 h in the absence or presence of MG132 (20 μM) to inhibit proteasome-mediated survivin degradation, followed by survivin immunoblotting. β-tubulin served as the equal loading control. *: *p* < 0.05; **: *p* < 0.01; ***: *p* < 0.001.

**Figure 3 ijms-21-01773-f003:**
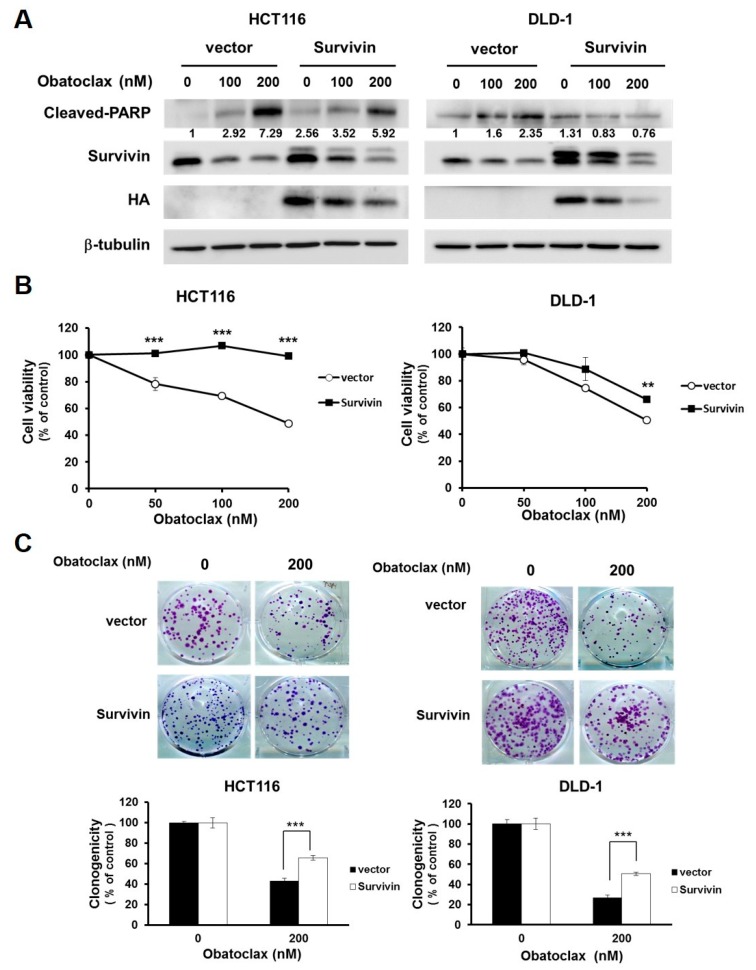
Survivin downregulation is required for the anti-CRC effect of Obatoclax. (**A**) Ectopic expression of survivin mitigates Obatoclax-induced CRC cell apoptosis. DLD-1 and HCT 116 cells stably expressing HA-tagged survivin and their respective vector control clones were treated with Obatoclax (0, 100, 200 nM) for 24 h, followed by immunoblotting for the levels of c-PARP, survivin, and HA. β-tubulin was used as the equal loading control. (**B**) Survivin overexpression attenuates Obatoclax-induced cytotoxicity. DLD-1 and HCT 116 HA-survivin stable clones and their respective vector controls were treated with Obatoclax (0, 50, 100, 200 nM) for 24 h, followed by MTS assay to evaluate cell viability. (**C**) Ectopic survivin expression sabotages Obatoclax-induced suppression of clonogenicity. DLD-1 and HCT 116 HA-survivin stable clones and their respective vector controls were treated with Obatoclax (0, 200 nM) for 24 h, followed by colony formation analysis. **: *p* < 0.01; ***: *p* < 0.001.

**Figure 4 ijms-21-01773-f004:**
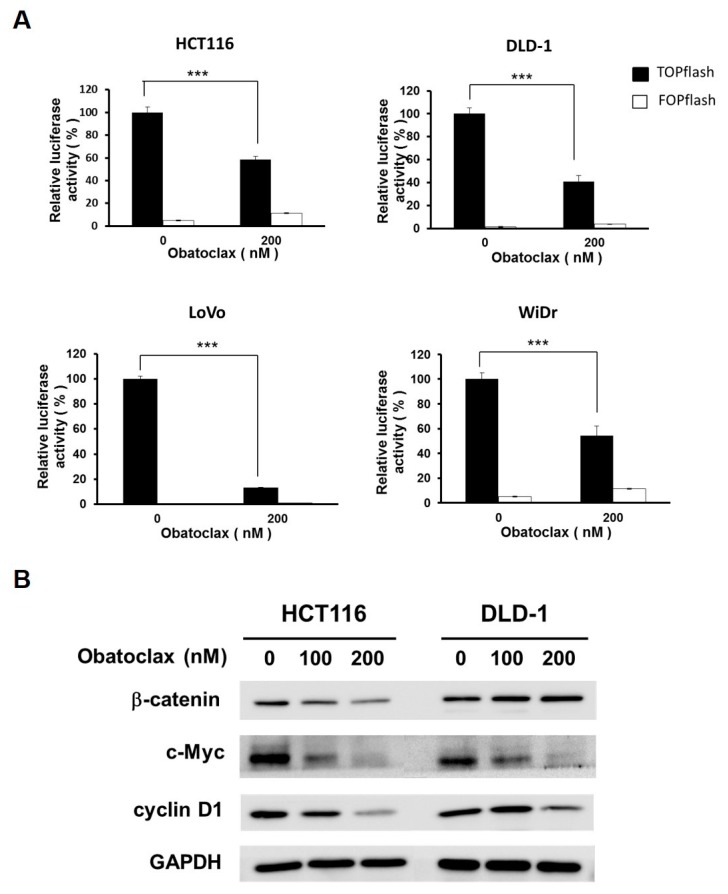
Obatoclax antagonizes WNT/β-catenin signaling. (**A**) Obatoclax decreases β-catenin-mediated TCF/LEF-dependent transcriptional activity. CRC cell lines were transiently transfected with M50 Super 8x TOPFlash (TOPflash), a luciferase reporter plasmid for the transcriptional activity of TCF/LEF, followed by 200 nM of Obatoclax treatment and ensuing luciferase activity assay. M51 Super 8x FOPFlash (FOPflash), a TOPFlash mutant, was used as a negative control. ***: *p* < 0.001. (**B**) Obatoclax downregulates WNT/β-catenin target genes c-MYC and cyclin D1. DLD-1 and HCT 116 cells were treated with Obatoclax (0, 100, 200 nM), followed by immunoblotting for the levels of c-MYC and cyclin D1. GAPDH was used as a control for equal loading.

**Figure 5 ijms-21-01773-f005:**
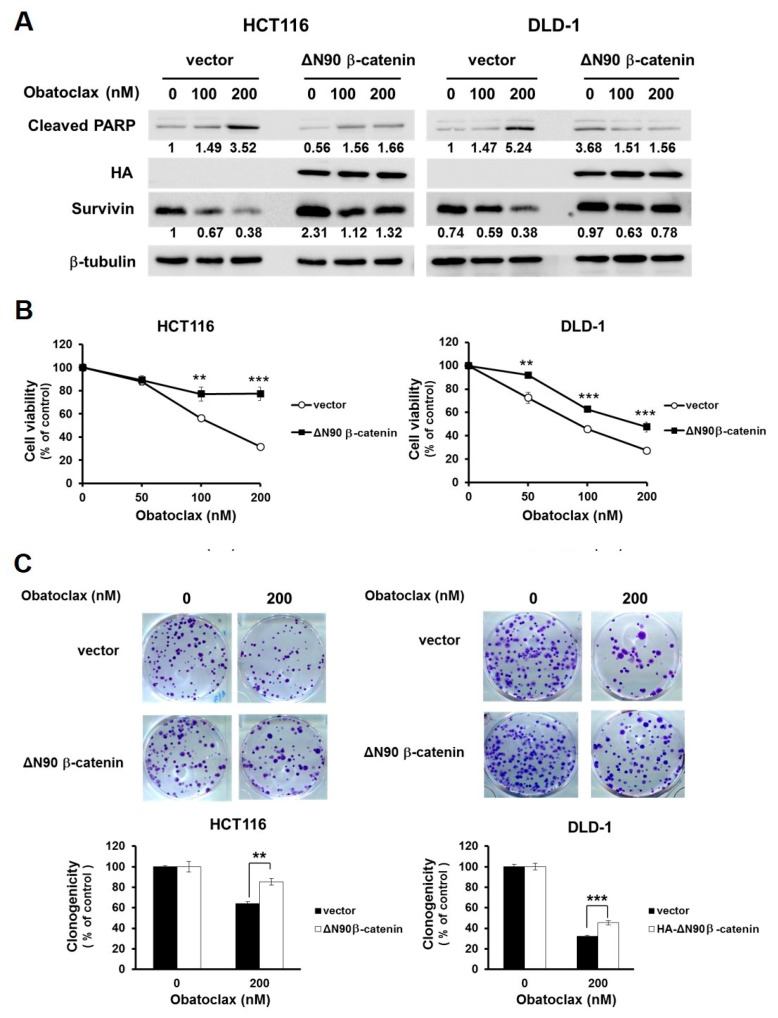
Blockade of WNT/β-catenin signaling is pivotal to the anti-CRC effect of Obatoclax. (**A**) Constitutive activation of β-catenin sustains survivin expression and confers resistance to Obatoclax-induced CRC cell apoptosis. DLD-1 and HCT 116 cells stably expressing HA-tagged dominant-active β-catenin (∆N90-β-catenin) and their corresponding vector control clones were treated with Obatoclax (0, 100, 200 nM) for 24 h, followed by immunoblotting for the levels of c-PARP, survivin, and HA. β-tubulin was used as the control for equal loading. (**B**) Constitutive activation of β-catenin attenuates Obatoclax-induced cytotoxicity. DLD-1 and HCT 116 HA-∆N90-β-catenin stable clones and their corresponding vector controls were treated with Obatoclax (0, 50, 100,200 nM) for 24 h, followed by MTS assay to evaluate cell viability. (**C**) ∆N90-β-catenin overexpression mitigates Obatoclax-evoked inhibition of clonogenicity. DLD-1 and HCT 116 HA-∆N90-β-catenin stable clones and their corresponding vector controls were treated with Obatoclax (0, 200 nM) for 24 h, followed by clonogenicity assay. **: *p* < 0.01; ***: *p* < 0.001.

**Figure 6 ijms-21-01773-f006:**
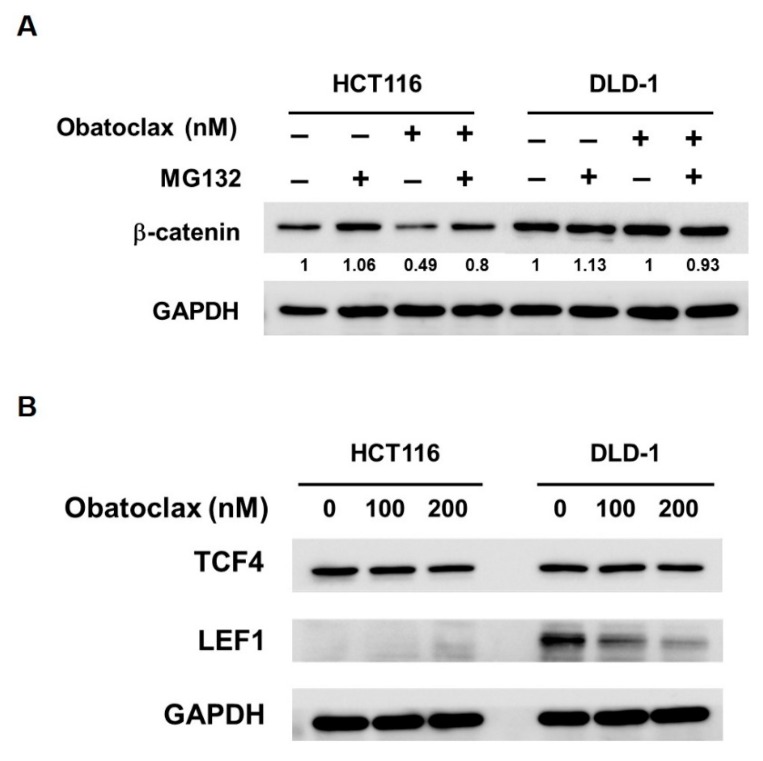
How Obatoclax antagonizes WNT/β-catenin signaling depends on the cell context. (**A**) Obatoclax induces proteasomal degradation of β-catenin in HCT 116 cells but not DLD-1 cells. CRC cells were treated with Obatoclax (200 nM) for 24 h without or with 2 h co-treatment of MG132 (20 μM). (**B**) Obatoclax downregulates LEF1 in DLD-1 cells. HCT 116 and DLD-1 cells were treated with Obatoclax (0, 100, 200 nM) for 24 h, followed by immunoblotting for the levels of TCF4 and LEF1. GAPDH levels were used as the equal loading controls.

**Figure 7 ijms-21-01773-f007:**
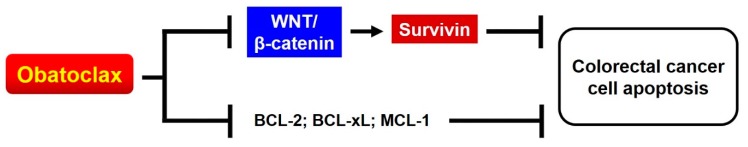
Schematic depiction of the mechanisms underlying the anti-CRC effect of Obatoclax, which likely involves the suppression of WNT/β-catenin–survivin signaling axis in addition to inhibiting the activity of antiapoptotic BCL-2 proteins.

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
