# Peer review of "Obatoclax, a Pan-BCL-2 Inhibitor, Downregulates Survivin to Induce Apoptosis in Human Colorectal Carcinoma Cells Via Suppressing WNT/β-catenin Signaling"

_ijms, 2020, doi:10.3390/ijms21051773_

Round 1
Reviewer 1 Report
In this study, the authors reveal that Obatoclax reduces the cell viability in the four CRC cell lines mainly mainly by stimulating cell apoptosis (enhance of activation of PARP, caspase8, caspase 9 and caspase 3) and inhibiting Wnt/β-catenin signaling-mediated surviving expression. The results are very clear and complete. However, I have five opinions to suggest as follow,
1.In Figure 1C, I suggest the authors to show the results of total form of PARP, caspase 8, caspase 9 and caspase 3 for demonstrate that Obatoclax could not affect the protein levels of these four factors
- It needs to list the reference about the transcription of the surviving-encoding gene, BIRC5.
- In Figure 3, how to explain that the expression of survivin was still decreased (Fig 3A, left panel), but not affect the cell viability (Fig 3B, left panel) after 200 nM Obatoclax treatment in HA-tagged survivin group in HCT116 cells.
- In the Reference 6 which the authors cited in the manuscript described that nuclear β–catenin cooperates with TCF/LEF to activate the expression of Wnt/β-catenin signaling target genes. (Cheng et al., 2019, page 475). In the result of Figure 6, the authors indicate that Obatoclax promotes β-catenin protein destabilization to block Wnt/β-catenin signaling. However, I suggest that the authors need to observe the difference of β-catenin expression in the nuclear and cytoplasmic fractionation.
- In lines 43-44, the authors describe that either inducing β-catenin protein destabilization or LEF1 downregulation likely contribution to Obatoclax-induced suppression of Wnt/β-catenin signaling. However, I only see the different expression of LEF1 in DLD-1 cell after Obatoclax treatment, but not in the other three cell lines. Therefore, I suggest that the authors need to express the overall results (TCF4 and LEF1 expression) for these four CRC cell line in the Figures 6.
Reviewer 2 Report
The manuscript by Chi-Hung R Or and co-authrs reports experiments on the antiapoptotic effect of Obatoclax, a BCL-2 inhibitor.
Actually, it demonstrates that the antiapoptotic effect of that drug is mediated also via inhibiting the WNT/beta-catenin pathway, thus downregulating survivin. This is a novel finding, and the demonstration is supported by functional experiments rigorously conducted.
The manuscript is well written and straightforward to understand for readers, methods clearly described.
Minor points. Mainly spell check check is required, i.e.:
line 121, change PAPR into PARP
lines 204 through 210, change catenein into catenin
line 296, change prodigioins into prodigiosin
figure 7, change caner into cancer
Author Response
We are very grateful to the Reviewer's positive opinions of our research, and also highly indebted to the Reviewer for indicating misspelled words in our original manuscript. All of the typos pointed out by the Reviewer have been corrected and highlighted in bold-and-red fonts in the revised manuscript.
Round 2
Reviewer 1 Report
All the author's responses are reasonable, the manuscript in its current form is ready for publication.